# BDNF Spinal Overexpression after Spinal Cord Injury Partially Protects Soleus Neuromuscular Junction from Disintegration, Increasing VAChT and AChE Transcripts in Soleus but Not Tibialis Anterior Motoneurons

**DOI:** 10.3390/biomedicines10112851

**Published:** 2022-11-08

**Authors:** Anna Głowacka, Benjun Ji, Andrzej Antoni Szczepankiewicz, Małgorzata Skup, Olga Gajewska-Woźniak

**Affiliations:** Nencki Institute of Experimental Biology, Polish Academy of Sciences, 02-093 Warsaw, Poland

**Keywords:** acetylcholine, nicotinic ACh receptor, brain-derived neurotrophic factor, spinal cord transection, neuromuscular junction, extensor and flexor muscle of the ankle joint, Schwann cell

## Abstract

After spinal cord transection (SCT) the interaction between motoneurons (MNs) and muscle is impaired, due to reorganization of the spinal network after a loss of supraspinal inputs. Rats subjected to SCT, treated with intraspinal injection of a AAV-BDNF (brain-derived neurotrophic factor) construct, partially regained the ability to walk. The central effects of this treatment have been identified, but its impact at the neuromuscular junction (NMJ) has not been characterized. Here, we compared the ability of NMJ pre- and postsynaptic machinery in the ankle extensor (Sol) and flexor (TA) muscles to respond to intraspinal AAV-BDNF after SCT. The gene expression of cholinergic molecules (VAChT, ChAT, AChE, nAChR, mAChR) was investigated in tracer-identified, microdissected MN perikarya, and in muscle fibers with the use of qPCR. In the NMJs, a distribution of VAChT, nAChR and Schwann cells was studied by immunofluorescence, and of synaptic vesicles and membrane active zones by electron microscopy. We showed partial protection of the Sol NMJs from disintegration, and upregulation of the VAChT and AChE transcripts in the Sol, but not the TA MNs after spinal enrichment with BDNF. We propose that the observed discrepancy in response to BDNF treatment is an effect of difference in the TrkB expression setting BDNF responsiveness, and of BDNF demands in Sol and TA muscles.

## 1. Introduction

Acetylcholine (ACh) is a key molecule to maintain locomotion at each level of the motor unit. In the spinal cord, the cholinergic propriospinal network modulates central pattern generator neurons and motoneurons (MNs), acting on the latter through the M2 muscarinic receptors [1,2,3,4]. In addition, ACh produced in the MNs plays a role in recurrent inhibition by activating the Renshaw cells [5]. In the periphery, ACh acts as the primary excitatory neurotransmitter that directly controls muscle contraction by activating the ligand-gated, nicotinic acetylcholine receptors (nAChR) of the neuromuscular junction (NMJ).

Complete spinal cord transection (SCT) leads to a loss of motor ability, due to the disruption of supraspinal tracts and altered functioning of the preserved spinal network [6,7]. Due to the dysfunction of interneurons, there is a profound depletion of excitatory and inhibitory synaptic inputs to the MN as well [8,9,10]. The number and branching of MN dendrites decrease, causing changes in their receptivity [11,12]. From the second day after injury, demyelination and the retraction of axons were described [7,13]. The results of our group showed that SCT in rats reduced peripheral proprioceptive glutamatergic and V0_C_-interneuron-derived cholinergic innervation of the lumbar MNs. That was accompanied by changes in the density of perineuronal nets in the vicinity of these MNs [8,9,14]. Subsequent studies revealed that the time-course of changes in the number of inputs on the ankle extensor (soleus; Sol) and flexor (tibialis anterior; TA) MNs was different, indicating a different vulnerability of extensor and flexor circuits to spinal injury [8,15].

Brain-derived neurotrophic factor (BDNF), expressed in the spinal interneuronal network and in MNs, maintains their morphology and function, prevents atrophy, and modulates axonal regeneration and plasticity, acting via full length tropomyosin receptor kinase B (TrkB.FL). It is abundant in multiple cell populations in the spinal cord and supraspinal motor nuclei [16,17,18,19,20]. The adeno-associated viral vector (AAV) is effective in BDNF gene delivery that allows BDNF tissue levels to increase, and is beneficial for paraplegic rats with altered excitability of the MNs [10,21,22,23,24]. Our group and the Mendell group showed that after SCT, AAV-BDNF-treated rats perform alternative locomotor movements with body weight support and plantar foot placement on a moving treadmill, which was not achieved in non-treated rats [10,22,25]. These positive effects of spinal BDNF overexpression became visible in the second week after SCT and lasted for at least 7 weeks.

The BDNF/TrkB signaling pathway is instrumental for the stability of NMJ and proper neurotransmission [26,27,28]; it also contributes to skeletal muscle regeneration [29,30]. BDNF is expressed in an activity-dependent manner [28,31,32,33] and modulates ACh release after receptor binding, which activates the protein kinase C pathway and influences the ACh exocytotic machinery by phosphorylating the Munc-18-1 and SNAP-25 proteins [34]. On the other hand, TrkB.FL activation is under the control of mAChRs [35,36], indicating that cholinergic signaling and BDNF signaling work closely together at the NMJ.

Changes that occur in the spinal cord after injury also lead to a strong impairment of the interplay between the MNs and all the muscle fibers that are innervated in the respective muscle [37,38,39]. The motor endplate, devoid of proper signaling, starts to disassemble. A massive disassembly is particularly affecting to a subset of NMJs in the TA muscle two weeks after injury; in another ankle flexor, the extensor digitorum longus (EDL), the loss of nAChR in the NMJ was also reported [37,40]. Synapse disintegration two weeks post injury has also been found for the Sol muscle in our experiments [41].

These findings have set the main aim of our work, which was to determine whether intraspinal AAV-BDNF treatment of rats with SCT could counteract the post-lesion changes in cholinergic signaling in MNs and at the peripheral synapses. For that purpose, we investigated the gene expression and protein distribution of pre- and postsynaptic markers associated with vesicle packaging, ACh release, degradation, and receptivity to ACh at the second post-surgery week. We also aimed to answer the question of whether the flexor and extensor muscles responded differentially to spinal injury and treatment, and to identify structural correlates of the potential therapeutic action of BDNF. The latter question was raised in light of data showing that BDNF transported anterogradely in motor axons affected skeletal muscle, perisynaptic Schwann cells, and played a role in the activity-dependent modifications of neuromuscular transmission [28,42,43,44,45]. 

This study shows that Sol MNs and Sol muscle, in contrast to TA muscle, were highly susceptible to intraspinal BDNF treatment after SCT. Preliminary accounts of this work have been presented elsewhere [41,46].

## 2. Materials and Methods

### 2.1. Animals

Experiments were carried out on 53 adult male Wistar rats (between 10–12 weeks of age), weighing 290–350 g prior to surgeries. The rats were obtained from the Animal Facility of the Medical University of Białystok (Białystok, Poland) and housed at the Animal House of the Nencki Institute of Experimental Biology in Warsaw. They were housed in a 12 h light-dark cycle, with free access to water and pellet food, under standard humidity and temperature conditions, in groups of 4–6. After surgery, the rats were placed in individual cages. Experimental protocols involving animals, their surgery, and care were approved by the First Local Ethics Committee in Warsaw (511/2013 14 November 2013, 523/2018 20 March 2018, 950/2019 21 January 2020), in compliance with the guidelines of the Directive of 24 November 1986 (86/609/EEC), 22 September 2010 (2010/63/EU) and the Animal Protection Act of Poland (2017) on the protection of animals used for scientific purposes.

The rats were subdivided into three groups (Table 1): intact control (N = 17), lesioned with intraspinal injection of phosphate buffered saline with a pH of 7.4 (PBS) (SCT-PBS; N = 16), and lesioned with intraspinal injection of AAV-BDNF (SCT-BDNF; N = 16). The use of animals injected with PBS as a control for the AAV-BDNF injected group served to compare between studies in which PBS has also been used [10,47]. Additionally, four lesioned rats with intraspinal injection of AAV-EGFP (SCT-EGFP) were prepared as a control carrying AAV vector, for comparison of data consistency on the IF cholinergic signal in NMJs between AAV-EGFP- and PBS-injected groups.

### 2.2. Spinal Cord Transection

In 36 rats, a complete spinal cord transection (SCT) was performed at the Th10 level under aseptic conditions, as previously described [8,48]. Rats were anesthetized with isoflurane (Baxter, Lessines, Belgium, 1–2.5% in oxygen) by facemask. As a premedication, the animals received a subcutaneous injection of butorphanol analgesic (Butomidor, Richter Pharma, Wels, Austria; 3.3 mg/kg). The skin on the back was shaved, disinfected, and cut. The vertebrae were then exposed after scalpel incision and separation of the muscles, and laminectomies were performed at the levels of the Th9/10 and Th11/12 vertebrae. At the Th9/10 opening, corresponding to the Th11/12 spinal cord level, the dura was opened and lignocainum hydrochloricum (2% solution; Polfa Warszawa S.A., Poland) was applied to the surface of the spinal cord. Transection was performed with surgical scissors and the gap between stumps was enlarged by aspiration to approximately 0.5 mm. The accuracy of the procedure was carefully inspected using a surgical microscope (Seiler Colposcope 955 Seiler Precision Microscopes, St. Louis, MO, USA). The area was then washed with a warm (36 °C) 0.9% NaCl solution, dried with absorbable cellulose, and the tissues were closed with surgical sutures.

### 2.3. Intraspinal PBS, AAV-EGFP or AAV-BDNF Injection

Immediately after SCT, at the site of the Th11/12 laminectomy corresponding to the spinal level L1, the dura was opened and lignocainum hydrochloricum (2%; AstraZeneca AB, Sweden) was applied to the surface of the spinal cord. With the support of a surgical microscope, AAV-BDNF, AAV-EGFP, or PBS was administered into the spinal cord through a fine glass capillary, approximately 0.7 mm bilaterally from the midline, and 1 mm deep. One µL of viral particles solution (3 × 10^8^ AAV1/2-SYN-BDNF particles or 3 × 10^8^ AAV-1/2-SYN-EGFP particles) or sterile PBS was injected at a speed of 0.1 µL per minute with the SP101i syringe pump (WPI, Sarasota, FL, USA). AAV1/2-BDNF construct coding for BDNF under control of the hSyn promoter was obtained as described earlier [10]. The map of a construct, and the Western Blot image demonstrating synthesized and released pools of biologically active recombinant mBDNF from transduced hippocampal cells in vitro are shown in the Appendix A. In rats, injections were done within half an hour after spinal cord transection. Five minutes after injection, the capillary was removed. After careful inspection of the lesion area, the surrounding tissues were closed with surgical sutures. After surgery, 5 mL of 0.9% NaCl was injected subcutaneously.

### 2.4. Retrograde Labeling of MNs

Two weeks before the SCT, groups of rats assigned to laser microdissection (LMD) of MNs received intramuscular injections of fluorescence tracers: cholera toxin conjugated with Alexa Fluor 488 (0.1% solution in PBS, Molecular Probes, Eugene, OR, USA) was injected into the TA, and Fast Blue (1% aqueous solution, Dr. Illing Plastics GmbH, Groß-Umstadt, Germany) was injected into the Sol muscle, to label MNs that innervated the muscles of interest. The rats were anesthetized as described above. The area of skin that covered selected muscles of the hindlimb was shaved, disinfected, and cut. The biceps femoris muscle was incised and the underlying TA and Sol muscles were exposed. Fluorescence tracers were injected into the TA (20 µL) and Sol (5 µL). Injections were performed with a Hamilton microsyringe with a 22-gauge needle attached. After 5 min of tracer delivery, the needle was kept in the muscle for at least 3 min to avoid dye leakage and then slowly removed. The injection site was cleaned with a warm 0.9% NaCl solution, and the biceps femoris muscle and skin were sutured. 

### 2.5. Postsurgery Treatment

After surgery, the rats were placed in warm cages, covered with blankets, and inspected until they recovered from the effects of anesthesia. Subsequently, the rats were placed in individual home cages with easy access to pellets and water. The antibiotic Baytril (Enrofloxacinum 5 mg/kg; Bayer GmbH, Leverkusen, Germany) and the analgesic Tolfedine (Tolfenamic acid 4%, 4 mg/kg, sc; Vetoquinol SA, Lure Cedex, France) were administered daily for the first three (after tracer injection) and five (after SCT) postoperative days. The rats were inspected twice daily and their bladders were manually voided. No significant health problems were observed after spinalization, except occasional bladder bleeding during the initial days after surgery.

The group prepared for LMD received Baytril and Tolfedine after the first surgery (tracers’ injections), the antibiotic Sultridin (24% Sulfadiazine + Trimethoprim, 30 mg/kg, Norbrook, Ireland) and the analgesic Vetaflunix (Flunixin meglumine, 2.5 mg/kg; VET AGRO, Poland) after SCT in the paradigm described above. If necessary, plastic collars (Harvard Apparatus, Holliston, MA, USA) were placed around the necks of the animals to protect their wounds from being licked. The collars were usually removed after the first postoperative night.

### 2.6. Tissue Processing

Two weeks after the SCT, rats were anesthetized with isoflurane and then received a lethal dose of Morbital (pentobarbital, 120 mg/kg body weight, intraperitoneal; Biowet Puławy Ltd., Puławy, Poland). Then they were transcardially perfused with ice-cold saline (IF, LMD, qPCR groups) or 4% paraformaldehyde (TEM groups). Their vertebral columns were then excised, placed on ice, and the spinal cords were dissected. Spinal cords and selected hindlimb muscles (TA, Sol) intended for qPCR analysis were cut with a McIlwain tissue chopper (Ted Pella Inc., Redding, CA, USA) to 0.8mm transverse sections, frozen on dry ice, and stored at −80 °C. In the LMD group, spinal cord L3-6 segments (where TA and Sol MNs are located) were surrounded by Jung tissue freezing medium (Leica, Nussloch, Germany) and frozen at −80 °C until they were cut into longitudinal sections, as described [9]. The muscles intended for IF were fixed in 4% paraformaldehyde in 0.1 M PBS for 60 min at room temperature (RT), then moved to a cold PBS solution and stored at 4 °C.

### 2.7. Fluorescence Labeling of NMJ Pre- and Postsynaptic Components 

Single muscle fibers (~12 fibers per animal) were isolated under a magnifying glass. Free-floating fibers were washed in PBS for 1 h and incubated with a solution of 5% normal goat serum (NGS) in PBST (PBS with 0.05% Triton-X) for the next one hour at RT to block nonspecific labeling. To detect the presynaptic components of the NMJ and Schwann cells (SC), fibers were incubated overnight at 4 °C with anti-VAChT antibody that identified synaptic vesicles containing acetylcholine vesicular transporter (1:500, guinea pig AB1588, Millipore, Burlington, MA, USA), combined with anti-S100 antibody that identified SC (1:500, rabbit Z0311, DAKO, Carpinteria, CA, USA). Both primary antibodies were diluted in PBST. For confirmation of the distribution of motoneuronal fibers and terminal Schwann cells (tSC), antibodies recognizing neurofilament (1:200, mouse monoclonal, M0762 DAKO, Carpinteria, CA, USA) and periaxin (1:3000, rabbit, Peter Brophy, Centre for Neuroscience Research, University of Edinburgh, Edinburgh, UK) were used. Subsequently, the fibers were washed three times, for 5 min each, in PBS, and incubated for 1 h (at RT) with secondary antibodies. Alexa Fluor 647 goat anti-rabbit, Alexa Fluor 488 goat anti-guinea pig (1:500, A21246 and A11073, Life Technologies, Carlsbad, CA, USA), and Bungarotoxin conjugated to Alexa Fluor 555 (1:1000, B35451 Invitrogen, Carlsbad, CA, USA) were used to detect postsynaptic nicotinic acetylcholine receptors (nAChR). After three washes, the fibers were labeled for the nuclei with Hoechst dye (1:1000). After another three washes, fibers were mounted on glass slides (Menzel-Glaser, Braunschweig, Germany) and air-dried, then coverslipped (0.17 mm thick, Menzel-Glaser) with Mowiol 4–88 medium and kept in the dark at 4 °C until analysis. Prior to regular staining, a single versus a triple labeling was verified to test whether the simultaneous triple reaction did not affect antibody binding. Negative controls (primary antibody omitted) showed the absence of immunoreactivity (data not shown).

### 2.8. Confocal Imaging and Image Analysis

The images of the NMJ were captured with a Zeiss LSM 780 confocal microscope (Carl Zeiss, Jena, Germany) using PL APO 40× (1.4 NA) DIC oil-immersion objective. The z-stacks of the 16 bit images consisted of digital slices collected at 0.21 μm intervals with a pixel size of 0.069. Scanning was sequential with a 1.6 μs dwell time. Images were collected at constant exposure parameters for each of four channels detecting fluorescence labeling for VAChT, S-100, Bungarotoxin, and Hoechst with the use of the 561 nm diode-pumped solid-state laser, the 633 nm helium neon laser, and the 488 nm argon laser.

The z-stacks of the images were subjected to a deconvolution procedure using Huygens Professional software (Scientific Volume Imaging, Hilversum, The Netherlands) to reduce the image distortion arising from light scattering. Based on a random set of images, an experimentally blind rater determined the parameters of the deconvolution algorithm to obtain maximum resolution for each antibody. The level of background and signal to noise ratio was selected for each channel independently; the number of iterations for the deconvolution algorithm was 40.

The deconvoluted images were randomly coded. The z-stacks of the digital images allowed for 3D object reconstruction (voxel size x/y/z of 0.065 × 0.065 × 0.21 μm^3^) and NMJ analysis was done using Imaris 7.6.1 software (Bitplane, South Windsor, CT, USA). The area and volume of the presynaptic and postsynaptic parts of the NMJ (illustrated in Figure 1) were measured, and the data were imported into Microsoft Excel (Microsoft, Redmond, WA, USA) for statistical analysis. All images were also visually examined; any change in the NMJ morphology or symptoms of nerve retraction were carefully noted. Additional analysis was done using ImageJ software for en-face captured NMJs. The percentage of area occupied by pre- and postsynaptic markers overlapping in maximal intensity projection (MIP) was examined, following the protocol of Jones et al. [49].

### 2.9. Localization of NMJs for TEM Imaging

Single muscle fibers were isolated under a magnifying glass. The fibers were then incubated with α-BTx-AlexaFluor™ 555, 1:1000 solution in PBS, for 15 min, and washed three times in PBS. NMJs were localized under Nikon Eclipse 80i microscope using 10× (0.30 N.A. DICL/N1) and 20× (0.50 N.A. DIC M/N2) objectives, and the part of the fibers containing NMJs was cut out with a scalpel under a magnifying glass to obtain ca. 1 mm × 1 mm samples for electron microscopy (TEM).

### 2.10. TEM Imaging of NMJ

For TEM purposes, fiber samples were postfixed in 2.5% glutaraldehyde (EM grade) in a 0.1 M cacodylate buffer (CB) for 1 h at RT, washed 3 × 10 min in CB, then postfixed in 1% osmium tetroxide for 1 h in CB at RT and washed in water (3 × 10 min). Next, the samples were dehydrated by incubation in increasing ethanol concentrations (50%, 70%, 90%, 96% and 2 × 100%; 10 min each) and cleared in a 1:1 mixture of ethanol + propylene oxide (1 × 10 min) and in 100% propylene oxide (2 × 15 min). During dehydration step with 70% ethanol, the samples were stained with 1% uranyl acetate (for 40 min). Finally, tissue was embedded in the Epon epoxy embedding medium (Serva). Polymerization was done at 60 °C for two days. Ultrathin sections (60 nm) were obtained using Leica Ultracut R ultramicrotome, collected on TEM slot grids of 1 mm × 2 mm (Ted Pella Inc., Redding, CA, USA). If preliminary examination did not reveal the presence of the NMJ, the sample was cut deeper until the desired structures were obtained. Sections were then post-stained with uranyl acetate and Reynold’s lead citrate. Electron micrographs were obtained with a Morada camera and a JEM 1400 transmission electron microscope at 80 kV (JEOL Co., Tokyo, Japan) in the Laboratory of Electron Microscopy, at the Nencki Institute of Experimental Biology, in Warsaw, Poland.

### 2.11. TEM Image Analysis

The number and size of synaptic vesicles (SVs) were measured following the protocol of Kim et al., using the cell counter function of ImageJ software [50]. The number of SVs analyzed in the TA NMJs amounted to: Control: 400, SCT-PBS: 320, SCT-BDNF: 320; in Sol NMJs: Control: 320, SCT-PBS: 190, SCT-BDNF: 360. Only SVs with a clearly distinguished membrane were selected for measurements of the outside diameter. A synaptic bouton area of 0.6–1 μm^2^ was used for calculation of the density parameter of the SVs. The density of the SVs was measured in the field of at least 500 nm^2^ remaining after exclusion of the area occupied by the axonal mitochondria. 

For measurements of the width of synaptic clefts and for the counting of active zones (AZ) only synapses with preserved pre- and postsynaptic membranes were taken into account. The selection of images was based on the criterion of the smallest visible length of the presynaptic membrane equal to 1 um/image. AZs were identified as electron-dense material aligned with postsynaptic junctional folds, as described in the literature [51,52]. The number of AZs was calculated manually using ImageJ software. Synaptic cleft distance was a manually measured inner width between the presynaptic membrane and postsynaptic crests of junctional folds. The average value of five measurements at different sites in each image was taken for further analysis. A total of 8 to 24 images/group were analyzed. Sequential screenshots of the ImageJ TEM analysis are shown in Appendix A.

### 2.12. Isolation of MNs Using Laser Microdissection (LMD)

Microdissection procedures were the same as those previously described in detail [9]. Briefly, spinal cord specimens containing L3-6 segments surrounded by Jung tissue-freezing medium were cut into 25 μm thick longitudinal horizontal sections using a cryostat (Leica CM1850). The sections were mounted on RNase-free PET-Membrane Frame Slides (Leica, Wetzlar, Germany, cat. no 11505190), which were immediately placed in a dry ice-precooled box and stored at −80 °C until processing (up to several weeks). Before LMD, the sections were dehydrated at increasing concentrations of ethanol solutions (up to 100%) and xylene. Single slides were placed in the Leica Laser Microdissection System (LMD 7000) to extract the TA and Sol MNs labeled with tracers. The MNs were identified, selected, and then microdissected with a UV laser under the microscope (objectives ×10, 0.32 NA and ×63, 0.7 NA). The dissected MNs were dropped by gravity into RNase-free tubes filled with the extraction buffer, and then incubated for 30 min at 42 °C to lyse the cells, which were centrifuged and stored at −80 °C. The total MN RNA was isolated using an Arcturus PicoPure™ RNA Isolation Kit (Applied Biosystems, Waltham, MA, USA, cat no. KIT0204) according to the manufacturer’s protocol. Next, the RNA was preamplified and reverse transcribed using a QuantiTect^®^ Whole Transcriptome Kit (Qiagen, Hilden, Germany, cat no. 207043, 207045). The concentration of cDNA was determined using a Quant-iT™ PicoGreen™ dsDNA Assay Kit (Thermo Fisher Scientific, Waltham, MA, USA, cat no. P7589).

### 2.13. mRNA Isolation, Transcription, and Analysis of Gene Expression by qPCR

RNA was isolated from whole spinal cord segments, microdissected MNs, and muscle samples, using the ZR RNA MiniPrepTM kit (Zymo Research Corporation, Irvine, CA, USA) according to the manufacturer’s protocol. An optional DNAse I digestion step was performed (Roche Applied Science, Indianapolis, IN, USA). The Transcriptor First Strand cDNA Synthesis Kit with random hexamer primers (Roche Applied Science) was used to convert the total RNA (0.5–1 µg) into coding DNA (cDNA). Reverse transcription was performed at 50 °C for 60 min followed by enzyme denaturation at 85 °C for 5 min. 

Quantification of the Bdnf, Ntrk2, ChAT, VAChT, nAChR, Chrm2 & Chrm4, Ache, S100 beta and MBP gene transcripts was performed using TaqMan hydrolysis probes and the LightCycler^®^96 sequence detection system (Roche Applied Science). Target-specific probes and forward and reverse primers were used as designed by the Universal ProbeLibrary Assay Design Center (Appendix A). Duplex qPCR was performed: each target gene transcript was analyzed in parallel with a control gene transcript of glyceraldehyde-3-phosphate dehydrogenase (GAPDH), with the use of cDNA obtained from the transcription of 50 ng of total RNA. Oligonucleotides were synthesized by the Laboratory of DNA Sequencing and Oligonucleotide Synthesis, at the Institute of Biochemistry and Biophysics, in Warsaw, Poland. The following thermal cycling profile was applied: preincubation at 95 °C for 10 min, denaturation at 95 °C for 10 s, annealing at 60 °C for 10 s, and extension at 72 °C for 10 s (fluorescence acquisition step). A total of 45 to 55 cycles were repeated. LightCycler^®^96 SW 1.1 software and the 2^−ΔΔCt^ method were used as a relative quantification approach for data analysis, based on the target and reference genes’ Ct (cycle threshold) value, as well as the cycle number at which the amplification curve reaches the threshold line.

### 2.14. Statistical Methods

The mean and standard deviation (SD) or standard error of the mean (SEM) were calculated for each data set. The Grubbs test of outliers was carried out for all data, and outliers were discarded (https://www.graphpad.com/quickcalcs/Grubbs1.cfm, accessed on 20 June 2022). Statistical significance was accepted at *p* ≤ 0.05. The Shapiro–Wilk test was used to verify the normality of data distribution, and the Levene’s test was used to verify the homogeneity of variance in the groups. One-way ANOVA, followed by the generalization of HSD Tukey (Spjotvoll & Stoline) were applied for unequal sample sizes. Because the assumption of the normality of distribution or homogeneity of variances was violated in some experimental groups, the non-parametric Mann–Whitney *U*-test was used for the comparison of independent samples (between groups), and the *t* test and Wilcoxon test were used for the comparison of dependent samples within groups (TA vs. Sol MNs and TA vs. Sol muscle specimens). The STATISTICA 13.1 software (StatSoft Inc., Tulsa, OK, USA) was used to analyze the data.

## 3. Results

### 3.1. Spinal Injection of AAV-BDNF Increases BDNF Expression in the Lumbar Segments and Sol Muscle Two Weeks after Spinal Cord Transection

BDNF transcript levels were examined by a qPCR assay in samples of mRNA extracts from the (1) lumbar spinal segments; (2) tibialis anterior (TA) and soleus (Sol) motoneurons; and (3) muscles isolated from the Control, SCT-PBS, and SCT-BDNF rats.

Figure 2A shows that in the Control group, BDNF mRNA expression in the low lumbar segments (L3-6) was high (mean = 3.5 × 10^−3^ BDNF/ref gene), which was in agreement with our previous results [10,53]. The expression of BDNF in tracer-identified TA and Sol MNs, located in these segments, was much lower than in the remaining tissue. Out of nine rats, it was above the detection limit in four rats only, in a range of 1 × 10^−9^–5 × 10^−8^ (Figure 2A). Despite that low transcript level, BDNF protein was clearly detected in these neurons (Appendix AA, control panel), and in the strongly immunopositive, dense mesh of processes surrounding them, which was in agreement with our earlier observations [48,54].

Two weeks after the SCT, the level of spinal BDNF expression in the SCT-PBS group did not decrease, while in the TA and Sol MNs, it was undetectable in the majority of rats (Figure 2A, left and middle panels).

AAV-BDNF administration to the L1-2 segment resulted in a 25-fold increase in BDNF mRNA in the L3-6 segments (Figure 2A, left panel; significant at *p* = 0.0043 vs. SCT-PBS and *p* = 0.0079 vs. Control group, Mann–Whitney *U*-test), and in a greater than 200-fold increase in the transgene-injected segment (not shown), confirming the effectiveness of the model. AAV-BDNF administration also resulted in high levels of BDNF mRNA in the TA MNs of four rats and in the Sol MNs of two rats; in the remaining rats, no overexpression of BDNF was found (Figure 2A, middle panel). 

These results are well reflected by the pattern of labeling revealed in parallel assays for a c-Myc tag from the BDNF transgene, which proved the presence of BDNF transgenic protein in the fibers of the lumbar spinal network, in the vicinity of the Sol and TA MNs, but not in their perikarya (Appendix AA). These observations allowed us to conclude that a mainly extra-motoneuronal pool of BDNF drove the changes in AAV-BDNF-treated animals at that post-lesion time, which was different from 5–6 weeks after the lesion, when BDNF transgenic protein was found to also be produced in multiple lumbar MNs [10].

Neither spinalization nor spinal overexpression of the BDNF gene significantly affected the mRNA levels of the high-affinity BDNF receptors (TrkB) assayed in the L3-6 segments (Figure 2B); in the SCT-PBS group, some animals responded with increased level of transcripts. The maintained potential of the L3-6 MNs to respond to the BDNF was indicated by the pattern of TrkB protein labeling in the MNs in all experimental groups, illustrated on the Appendix AB.

We also asked whether BDNF overexpression affected the levels of p75 neurotrophin receptor, which binds all neurotrophins with equal affinity. A remarkable increase in the intensity of p75 labeling was found in the SCT-PBS group, and a weaker signal was observed in the SCT-BDNF group (Appendix AB). 

In the isolated pools of the TA and Sol MNs, spinalization caused differential effects. In the TA MNs, the levels of TrkB mRNA decreased three times, and was further reduced by AAV-BDNF treatment, while in the Sol MNs, the levels of TrkB mRNA did not decrease in spinal groups (Figure 2B, right panel). These results suggested that most of the spinal cells maintained responsiveness to BDNF signaling after the lesion, however in the TA MN pool, TrkB downregulation may have impaired neurotrophic signaling to the TA muscles, which was further reduced by overexpressed BDNF.

In both muscles in the Control group, BDNF mRNA levels were lower than those in the spinal cords (Figure 2A). However, there was an approximately 50-fold difference between BDNF mRNA expression in the Sol (mean = 3.7 × 10^−4^) and in the TA (mean = 7.1 × 10^−6^) muscle (*p* = 0.008, Wilcoxon test), suggesting that, under physiological conditions, the TA muscle was a less abundant source of neurotrophic support and a less obvious target of autocrine regulation by BDNF than the Sol muscle. 

Spinalization led to a strong decrease (by more than 90%) in the levels of BDNF mRNA in the Sol (*p* = 0.0002, Mann–Whitney *U*-test) but not in the TA muscle. Surprisingly, spinal BDNF overexpression partially counteracted that deficit (SCT-PBS vs. SCT-BDNF group, *p* = 0.017, Control vs. SCT-BDNF group, *p* = 0.075, Mann–Whitney *U*-test). Furthermore, compared to the SCT-PBS rats, all SCT-BDNF rats revealed a higher expression of BDNF mRNA in the Sol muscle, while only two out of five SCT-BDNF rats revealed a higher expression of BDNF mRNA in the TA muscle (Figure 2A).

This set of data showed a clear difference between the TA and Sol motor circuits in several aspects of BDNF signaling. The disparity in the control abundance of TrkB mRNA in the MNs and BDNF mRNA in the muscles, together with differential responses to spinalization and BDNF-treatment, suggest that post-lesion maintenance of NMJ and cholinergic activity, if under the control of BDNF-TrkB signaling, can occur to varying degrees in both muscles.

Because muscle receptors and the nerve fibers that innervate them require NT-3 neurotrophin for recovery after damage [55,56], and the majority of neurons innervating muscle spindle receptors, as well as perisynaptic and myelinating Schwann cells, show expression of TrkC mRNA [57,58], we asked the question of whether spinal BDNF overexpression exerted an effect on the NT-3/TrkC neurotrophin system. We analyzed the effect of BDNF in the spinal cord and in the TA and Sol muscles. As shown in Appendix A, spinalization did not alter the expression of NT-3 or TrkC in the L3-6 spinal segments and in the muscles, but the BDNF had a stimulatory effect on both genes at the spinal level (* *p* ≤ 0.05). The analysis also revealed the disparity in the abundance of NT-3 mRNA and TrkC mRNA in the control muscles, pointing to higher neurotrophin dependence of the Sol circuit. 

### 3.2. The Effect of Spinal Cord Transection and AAV-BDNF Treatment on NMJ Integrity, Myelination of the Nerve Endings, S100b and Mbp Expression in Schwann Cells

To examine the effect of SCT and BDNF overexpression on the integrity of the nerve ending and the endplate, neuromuscular junctions (NMJs) with perfectly labeled structures selected from the triple-labeled TA (*n* = 79 NMJ) and Sol (*n* = 98 NMJ) muscles (Figure 3A,B). Each NMJ were reconstructed in three dimensions (3D) and subjected to blinded qualitative examination of the adjacency of IF signal identifying myelinated nerves that contacted the NMJ (Figure 3B). The experimenter analyzed NMJ images answering yes/no to the question: “Is a myelinated nerve fiber (immunolabeled with S-100) visible, and does it create a continuum with the tSCs that surround the endplate?”

The S-100 IF signal analysis of a continuity of the myelinated nerve fiber revealed a significant reduction in the number of proper NMJ-nerve contacts in both muscles after SCT (Figure 3C). BDNF overexpression resulted in the preservation of a higher number of contacts with normal morphology, which in the Sol muscle of the SCT-BDNF group was maintained at a level that did not differ significantly from the Controls. 

The pattern of periaxin staining, which shows a myelinating subpopulation of SCs, revealed that in animals subjected to spinal cord transection, periaxin staining did not cover terminal nerve branches and appeared to be more dispersed than in intact rats (Appendix A). In most denervated or demyelinated NMJs, and in both spinal groups, non-myelinating tSCs were still abutting the endplate, which was a phenomenon also reported by Burns et al. [37]. This result raised the next question on the molecular changes in the subpopulation of myelinating Schwann cells assisting the most distal parts of the nerve, achievable in the muscle, and in the accompanying, non-myelinating terminal Schwann cells. We found that S-100b expression characterizing both cell populations was significantly reduced; and was similar in each spinal group (Figure 3D), which suggested a loss of these cells after spinalization or their de-differentiation [59].

### 3.3. The Effect of Spinal Cord Transection and AAV-BDNF Treatment on Markers of Cholinergic Transmission in TA and Sol NMJ: Evaluation of the Endplate Integrity

The image analysis of the presynaptic part of the NMJs visualized by VAChT immunofluorescence revealed that a size of the axon terminal bouton area occupied by VAChT was different between the TA and Sol NMJs in the Control animals (Figure 4A). Sol MN terminal boutons were characterized by a greater area (1.4×) and volume (1.5×) than TA MN terminal boutons (*p* = 0.052 and *p* = 0.012, t test for dependent samples). The infolded postsynaptic membrane of the NMJs, visualized by means of fluorescent-tagged bungarotoxin (BTx) bound to nAChRs, showed a similar feature: Sol NMJs had a greater area (1.3×) of the junctional postsynaptic membrane occupied by nAChRs than TA NMJs (*p* = 0.009, t test for dependent samples). Endplates of predominantly slow twitch muscles of the rats were shown to be larger than in the fast twitch muscles [60,61].

Both the VAChT IF signal area and volume decreased significantly in the NMJs in both muscles in response to SCT (area: TA *p* = 0.0002, Sol *p* = 0.0023; volume: TA *p* = 0.0002, Sol *p* = 0.0002; ANOVA and Tukey’s post hoc test). A presynaptic part of the TA NMJ was more affected than that of the Sol NMJ; in each spinal group, a complete loss of the VAChT IF signal at terminals was more frequent in the TA MNs (Figure 4A; for comparison see Films 1–3 of Sol NMJs in Appendix A). On the postsynaptic side, spinalization did not change the area and volume of the BTx-labeled structures in the TA NMJs, while in the Sol NMJs, the volume of BTx-labeled structures decreased (Control vs. SCT-PBS *p* = 0.026, Control vs. SCT-BDNF *p* = 0.032). 

To examine the effect of SCT and BDNF overexpression on endplate integrity, 3D-reconstructed NMJs were subjected to a blinded qualitative examination of the adjacency of markers of pre- (VAChT) and postsynaptic (nAChR) cholinergic components of the NMJs from the side view (Figure 4D). The experimenter analyzed NMJ images answering yes/no to the question: “Is the VAChT IF in direct contact with the nAChR-associated Btx signal?” 

Analysis of spatial VAChT/BTx signal adjacency showed profound decreases in the SCT-PBS group (50–70% synapses lost contact between pre- and postsynaptic components). The decrease resulted primarily from the marked loss of the VAChT signal, and was deeper in the TA muscle (Figure 4B,D, Films 1–3 in Appendix A). Spinal BDNF overexpression prevented part of the Sol NMJs from synapse disintegration (SCT-PBS vs. SCT-BDNF *p* = 0.07, Mann–Whitney *U*-test), but did not prevent this process in the TA NMJs. Additional measurements of the VAChT/BTx-nAChR labeling overlap in the en-face images, with the use of maximal intensity projection analysis, confirmed a profound decrease inthe overlapping in spinal groups, both in the TA and Sol NMJs (Figure 4E). There was no difference in the degree of VAChT/BTx-nAChR overlap between spinal PBS and AAV-EGFP control groups. Thus, both types of analysis indicated that pre- and postsynaptic parts apposition was reduced after spinalization, but the side view analysis provided additional information on the change in possible direct contact.

### 3.4. The Effect of Spinal Cord Transection and AAV-BDNF Treatment on Density and Size of Synaptic Vesicles, a Number of Active Zones and Synaptic Cleft Distance 

Next, to further examine the effect of SCT and BDNF overexpression on the presynaptic and postsynaptic parts of the NMJ, we investigated its ultrastructure. For the measurements of the SVs, density, synaptic clefts, and counting of active zones in the synapses with preserved pre-and postsynaptic membranes were selected. The number, size, and frequency size distributions of synaptic vesicles (SVs), synaptic cleft distance, and the number of active zones per 1 µm of membrane were determined in the NMJs of the Control, SCT-PBS, and SCT-BDNF rats (Figure 5).

SCT did not change the mean density of the SVs in the populations of NMJs with maintained synaptic contacts in either the TA or Sol MN terminals, albeit in some cases in the Sol MNs, the density of the SVs was decreased, as exemplified in Figure 5. Also, in the Sol NMJs, SCT led to a reduction in a pool of large SVs, at the expense of smaller ones (Figure 5B, frequency distribution). After AAV-BDNF treatment, the mean diameter of the SVs was at a control level in both MN groups. The increased abundance in large vesicles in the TA NMJ after treatment with AAV-BDNF indicated a mechanism that led to the modulation of SV size, and possibly to the efficiency of neurotransmitter release. 

The average synaptic cleft distance in the Controls was 44.83 nm (SD = 5.52) in TA junctions and 52.8 (SD = 5.73) in Sol junctions. In the spinal groups, the distance was comparable and amounted to 51–52 in both NMJ types. The number of active zones was comparable between groups. We concluded that in the subpopulation of junctions with preserved morphology in all morphological parameters under study were maintained within the control range.

### 3.5. SCT and BDNF Treatment Change Expression Levels of Genes Coding for Vesicular Acetylcholine Transporter, Enzymes and Receptors Involved in Acetylcholine-Driven Neurotransmission 

Gene expression of the cholinergic signaling molecules was examined in the MNs and muscle tissue by qRT-PCR. In the MNs of the control rats, the expression levels of the vesicular acetylcholine transporter (VAChT) in the TA MNs were significantly higher than in the Sol MNs (*p* = 0.012, Wilcoxon test), suggesting that the ACh turnover was higher in the TA than in the Sol MN pool. The same relation was observed for choline acetyltransferase (ChAT) mRNA levels, but the result did not reach a statistical significance (*p* = 0.076, Wilcoxon test). Acetylcholinesterase (AChE) mRNA levels were comparable in both MNs types (Figure 6, upper panel). 

In the SCT-PBS group, the levels of VAChT mRNA and ChAT mRNA in the Sol MNs did not change, while the levels of AChE mRNA decreased significantly. This response altered a balance in expression between the enzymes involved in ACh synthesis and hydrolysis. The treatment with AAV-BDNF caused a strong up-regulation of the VAChT mRNA and the restoration of the AChE mRNA levels in the Sol MNs. The TA MNs responded to SCT differently: a moderate increase in ChAT mRNA expression was accompanied by a decrease in VAChT and AChE expression. The AAV-BDNF treatment did not normalize these changes and further reduced the AChE expression. 

In the control TA and Sol muscles, the levels of AChE mRNA expression were comparable (Figure 6, lower panel). However, they were more than two orders of magnitude lower than AChE expression in the MNs, suggesting that the regulation mechanisms and turnover of the presynaptic and postsynaptic AChE pools in the NMJs were different. 

The spinalization effect on the AChE mRNA differed between muscles. As shown in the Figure 6, lower panel, SCT caused a small increase in AChE mRNA in the TA muscle (*p* = 0.055, Mann–Whitney *U*-test) and a 4-fold increase in the Sol muscle (*p* = 0.000). After AAV-BDNF treatment, the levels of AChE transcripts did not differ from the controls in both muscles.

In the control TA and Sol muscles, the expression of postsynaptic nicotinic receptor nAChR alpha 1 subunit, which plays a role in ACh binding and channel gating, was differentiated, with much higher expression in the Sol muscles (*p* = 0.005, Wilcoxon test). We did not observe significant changes in the nAChR mRNA after SCT, but BDNF treatment resulted, to our surprise, in markedly lower expression of the nAChR in the Sol muscles.

Because our previous studies revealed that SCT led to a profound decrease in M2 receptor transcripts in MNs at 2 weeks, followed by reduced ligand binding to the mAChRs detected in the L5–L6 spinal segments [4,9] where the ankle extensor MNs are predominantly located, we also aimed to examine whether SCT and AAV-BDNF treatments affected M2 and M4 receptor expression at the level of the NMJs in the Sol and TA muscles. We assumed that M2/M4 expression levels would reflect levels of both muscular and axonal receptor mRNA, as the local translation of proteins in the axonal compartment of other systems has been well described [62,63], and of Schwann cells which express M1–M4 muscarinic receptor subtypes [64,65].

In the control muscles, the expression of NMJ muscarinic receptors M2 and M4 was low and highly correlated (TA: r = 0.910, *p* = 0.002; Sol: r = 0.816, *p* = 0.013). In the SCT-PBS group, no significant change in the transcript levels of either receptor was found in the muscles, but their correlation was markedly reduced. The AAV-BDNF treatment reduced the expression of M2 in the Sol muscles, but recovered to some extent a quantitative relationship between receptors (r = 0.869, *p* = 0.056) (Figure 6).

## 4. Discussion 

In this study, we showed a set of changes that pointed to a strong modulation of cholinergic signaling in the motor circuit of the Soleus muscle in response to spinal injury and BDNF treatment. We investigated the levels of expression of enzymes that catalyze acetylcholine synthesis and hydrolysis, and of the vesicular transporter of ACh responsible for packaging ACh into synaptic vesicles. A clear pattern of changes in the presynaptic component included a decrease in the AChE transcript after transection, and an opposite change in that transcript after AAV-BDNF treatment, which led to the normalization of AChE expression in the Sol MNs. BDNF treatment also led to the marked elevation of VAChT expression in the Sol MNs. 

Our study also pointed to a new possible mechanism contributing to the regulation of ACh release by BDNF. In both the TA and Sol muscles, AAV-BDNF treatment led to a decreased expression of M2 mAChR receptors. Because presynaptic M2 mAChRs have been reported to suppress ACh release [66,67], their reduced expression, if followed by a parallel change in their membrane abundance and function, might have resulted in higher ACh release and synapse efficiency. The ACh signaling may have additionally been increased in the Sol muscle, in which the BDNF normalized the muscular AChE transcript level, which was upregulated four times after SCT. The lower availability of AChE after BDNF treatment may have led to the extension of the lifetime of the ACh molecules in the synaptic cleft, which compensated for a decrease in nAChR transcripts, and consequently rescued muscle contractions. 

We also broadened the scope of evidence for different vulnerabilities of the TA and Sol circuits to spinal cord injury, as shown in our previous reports [8,9]. Based on the NMJ morphology analysis at 2 weeks post-lesion, we showed a higher vulnerability to SCT of the TA NMJs, and lack of TA NMJ responsiveness to BDNF treatment. In contrast, in the Sol muscle, BDNF partially preserved NMJ innervation. These findings add substantially to our understanding of regulation of slow and fast muscle activity, pointing to BDNF as a modulator of cholinergic signaling with target selectivity, making it potentially useful in therapy addressed to selected pools of MNs. Motoneuronal changes in the ChAT and VAChT mRNA levels, NMJ integrity, and presence of myelinating Schwann cells, in the context of the distribution and abundance of BDNF and TrkB transcripts in the spinal cord and muscles, are shown in Figure 7.

### 4.1. Differential Characteristics of Critical Components Regulating ACh Synaptic Availability in Sol and TA Motor Circuits of Control Rats 

A significantly lower level of VAChT transcripts in the Sol MNs, compared to the TA MNs detected in our study, was in line with the results by Woods and Slater, who reported lower ACh release in the Sol muscle, as compared to ACh release in another ankle joint flexor, the EDL [61]. This characteristic was in line with the data which show that fast and slow type muscles differentially responded to cholinergic deficits [68,69]. In our experiments, differences between the concentrations of cholinergic markers in the MNs, as well as in the TA and Sol muscles, were found only at the mRNA level. The VAChT protein abundance in the NMJs was comparable between the TA and Sol. The area of the endplate labeled by BTx was larger in the Sol muscle. Similar morphometric results were obtained by Woods and Slater [61], who found that Sol NMJs were 50% larger than those of the EDL, and by Mech et al. [70], who showed that fast-twitch fibers were more likely to be innervated by smaller neuromuscular junctions.

The level of expression of AChE in the TA and Sol MNs was similar. In the muscles, the AChE mRNA level in the TA was higher than in the Sol, which was in line with studies by Sketelj and coworkers who reported several-fold higher levels of AChE in fast compared to slow muscles [71].

### 4.2. Differential Expression Pattern of BDNF and Its TrkB Receptor in Motor Circuits of TA and Sol Muscles 

This study is the first to show a very low abundance of BDNF transcripts in the TA and Sol MNs in rats. These results added to our early observation of differential BDNF mRNA expression detected by means of *in situ* hybridization in multiple pools of MNs located in the L3–L4 spinal segments [72]. Importantly, there was a pool of MNs almost devoid of BDNF expression [73]; thus, the Sol and TA MNs may belong to this category. Since the BDNF protein is abundant in multiple MN pools and undergoes activity-dependent up-regulation [48,54,72] we assumed that posttranscriptional regulation played a crucial role in adjusting the BDNF levels in the MNs. 

An interesting aspect that emerged from our analysis of the TrkB mRNA expression was a significantly higher level of TrkB mRNA in the TA than in the Sol MNs. It stood in opposition to the early study by Copray and Kernell, where TrkB and BDNF mRNA expression in the Sol and EDL MNs did not show differences in a non-radioactive *in situ* hybridization experiment [74]. The difference found by us may point to higher responsiveness of the TA MNs than the Sol MNs to spinal BDNF, which could be associated with their different functions.

Despite the low level of BDNF expression in both types of muscles, there was a difference between them, with a much higher expression of BDNF in the Sol than in the TA muscle. Similarly, NT-3 and TrkC expression were also significantly higher in the Sol than in the TA muscle. That characteristics may reflect higher neurotrophic demands of the Sol circuit and may functionally relate to the daily duration of the Sol muscle activity, which in intact animals is approximately seven times longer than that of the TA muscle [75]. Our data were in line with recent work by Cefis et al. [76] who showed that the expression of BDNF and of phosphorylated TrkB receptors was higher in the Sol than in the medial head of the gastrocnemius muscle (which contains more fast-twitch fibers than the Sol). Interestingly, at the same time the BDNF was higher in type II fibers, which prevail in fast muscles, than in type I fibers. Such a contradiction may be explained by the much higher abundance of the BDNF in the Sol muscle, owing to BDNF synthesis by the endothelial cells. These cells were more abundant in Sol, due to the rich network of capillaries in that muscle; thus, the higher expression of BDNF may not be related primarily to its expression in myocytes [76]. 

### 4.3. Disparity of BDNF and TrkB mRNA Regulation Patterns in Response to SCT in the TA and Sol Motor Circuits 

The analysis of gene expression did not show significant differences in the levels of motoneuronal BDNF mRNA in the SCT-group compared to Control group for both types of MNs. This was in line with studies reporting that a low level of BDNF mRNA observed in control MNs was not affected by SCT [73,77]. In contrast, the TrkB mRNA level in the MNs underwent strong changes after SCT in the TA MNs, while in the Sol MNs they remained at the control level. 

In the muscles, SCT only had an effect on the level of Sol BDNF expression, and significantly decreased it. Considering the activity-dependent expression of BDNF, it was in line with the different states of those two muscles in paraplegic rats, as reported by us previously. In the Soleus muscle, which is constantly relaxed when the animal drags the paws on the dorsal aspect, the effect of disuse is more pronounced [8].

### 4.4. Significant Increase in BDNF mRNA in Sol Muscle but Not TA Muscle after AAV-BDNF Intraspinal Administration

The intraspinal injection of AAV-BDNF led to a high overexpression of BDNF in the L3–L6 segments of the spinal cord. The general capacity of the spinal system to respond to the treatment was confirmed by the analysis of TrkB gene expression, which showed preserved receptivity to the BDNF. No observed increase in lumbar expression of the TrkB at 2 weeks after SCT was contrary to what has been documented by Keeler and coworkers [10,77]. We did not notice a decrease in the TrkB expression level after AAV-BDNF treatment either, which was reported by us at 5 weeks after SCT [10]. 

The treatment led to an increase in BDNF mRNA in the MNs innervating the TA. Only one rat from the SCT-BDNF group overexpressed BDNF in the Sol MNs. This was an expected result, considering that the TA MNs were located more proximal to the site of injection as compared to the Sol MNs, thus increasing the probability of neuronal transfection by the AAV-BDNF [78]. 

Few studies have explored the motoneuronal expression of BDNF after the intraspinal injection of AAV-BDNF. According to the previous result of our group [10], six weeks after injury, the exogenous BDNF protein was detected in large neurons of the L2 ventral horn, while in lower lumbar segments, only inputs from BDNF expressing projections were detectable on large neurons. It was in line with the report of Boyce and colleagues for the same time point after the administration of AAV-BDNF below the site of the injury. Transfected MNs were visible at the level of the L2 spinal cord, but in the L5 segment, only positive fibers were observed [22].

In a current study, after the intraspinal administration of AAV-BDNF, the level of BDNF mRNA in the Sol muscle increased, and tended to normalize. In the TA, no changes were observed. The source of that increased production in the Sol needs further investigation. We assumed it was not an effect of the axonal transport of transgene mRNA, as we didn’t observe any increase in the motoneuronal expression of BDNF in the Sol muscle. We hypothesize that the BDNF is expressed in muscle fibers during functional improvement, which was observed in this study, since the BDNF is target-derived and expressed in an activity-dependent manner [28,31,32,79]. Its higher production in Soleus muscle can be considered as an effect of higher basal neurotrophic demands of the Sol muscle. A second possibility is that the AAV-BDNF spreads through the blood system. The expression of BDNF in the blood vessels was proposed by Cefis et al. [76], and should be taken into consideration in our experimental model, as we used the synapsin promoter to express BDNF, and synapsin-1 has also been reported to be expressed in the epithelial cells of blood vessels [80]. 

### 4.5. Presynaptic and Postsynaptic Changes in Cholinergic Components in the TA and Sol NMJs after SCT and BDNF Treatment

A decrease in VAChT mRNA was detected in the TA, but not in the Sol MNs at 2 weeks after SCT, and may have referred to data described by Magalhães-Gomes, who examined skeletal muscles in mice with reduced expression of VAChT in MNs, in which fast- and slow-type muscles responded differentially to cholinergic deficits. In the TA circuit, a deficiency led to muscle fiber atrophy, while in the Sol circuit, it led to muscle fiber hypertrophy [68]. A rich diversity of gene expression beyond myosin isoforms, which defines the characteristics of any muscle fiber, was found in the Sol and EDL muscles [81]. This diversity may underly the range of muscle phenotypes formed in response to differentially altered cholinergic signaling. That alteration in our experiments also involved the level of ChAT, which increased significantly in the TA MNs, while no change was observed in the Sol MNs. The ChAT response may be a compensatory mechanism, as the VAChT transcript level only decreased in the TA MNs, and was accompanied by a pronounced decrease in VAChT protein in synaptic terminals in the NMJ. However, a proposed differential regulation of VAChT vs. ChAT gene expression requires further studies that consider the data on the location of gene encoding of the VAChT within the first intron of the gene encoding ChAT [82]. This arrangement implies that these two genes, whose products are required for the expression of the cholinergic phenotype, are coregulated, as shown by many factors [83]. Some data have shown that these two genes can be regulated synergistically or differentially, depending on the activated signaling pathways [84]. 

In previous sections, we discussed a clear difference between the TA and Sol motor circuits in several aspects of BDNF signaling in the control and SCT conditions. Therefore, post-lesion maintenance of the NMJs and of the cholinergic activity, which are regulated by BDNF-TrkB pathway, may have occurred to varying degrees in both muscles. In general, the TA NMJs were more vulnerable to SCT than the Sol NMJs, as shown by examination of their morphology and synaptic connectivity; our observation was consistent with the result of Burns et al., who also reported destabilized synapses in the TA, but not in the Sol [37]. Postsynaptically, we found no changes in the volume or area occupied by a ligand-binding population of nAChRs in the TA NMJs, and a decrease in the Sol NMJs. These observations did not confirm with those by Gronert et al. [85], who reported that any immobilization of the skeletal muscle resulted in up-regulation of the nAChRs in the endplate. Our data added to the reported variability of NMJs’ response to injury [37,86].

The TEM analysis revealed no significant differences between the TA and Sol NMJs, and no clear effect of SCT or BDNF-treatment on the SVs pool, synaptic cleft distance, and number of active zones. However, it should be noted that this set of data reflects the state of the SVs in the NMJ subpopulation with maintained integrity, because NMJ sections showing dispersed presynaptic structure and membrane fragmentation in the TEM were eliminated from counting and measurements. Thus, we discuss the SVs pools in these NMJs, which were preserved. Non-compliance between changes in the VAChT protein level and the quantity of SVs may also suggest the presence of non-functional or empty vesicles in rats after transection. That is in line with studies by Rodrigues and coworkers on mice with a 70% reduction in VAChT expression, who showed that the number of vesicles in the NMJs were the same compared to WT mice, and only a shape and distribution of the SVs in response to electrical stimulation differed [87]. Interestingly, Prado et al. described no changes in the neuromuscular phenotype in knockdown mice with a 45% reduction in VAChT expression (heterozygotes), but a 65% reduction (homozygotes) resulted in the significant deficiency of neuromuscular function and physical capacity [88]. These findings suggest that in our model, where SCT caused a massive loss in VAChT staining in NMJ terminals and BDNF treatment resulted in great functional improvement, the BDNF-related mechanisms rescued ACh release and neuromuscular transmission.

### 4.6. Possible Mechanisms of Modulation of Cholinergic Signaling by BDNF 

Taking into account different effects that the SCT and BDNF treatment had on cholinergic signaling in slow and fast muscles, the search for a possible mechanism of action of BDNF should start with the characteristics of processes that differentiate the functioning of the slow and fast motor unit. It was shown that, at 1–2 months after SCT, the slow to fast fiber transition of the paralyzed muscles began [89,90], and BDNF overexpression was shown to promote this change [91]; therefore, revealing possible mechanisms in our model of very high BDNF overexpression becomes challenging. 

We propose that after SCT, the intraspinal injection of AAV-BDNF acted on the cholinergic system at two levels. At first, transfected interneurons, enriched in BDNF, released neurotrophin that stimulated MNs in the lumbar spinal segments. We expected this mechanism to operate mostly in Soleus MNs, due to their high receptivity to BDNF, in contrast to TA MNs, which are almost devoid of receptors. As a result, an increase in ChAT expression was induced, which was shown to be strongly correlated with locomotor performance after spinal injury [92], with a similar increase in VAChT expression and normalization of AChE synthesis. The second level could be a direct action of BDNF at the NMJ through presynaptic TrkB receptors that stimulate neurotransmitter release [27,28]. Under control conditions, BDNF is available for this process, due to its activity-dependent secretion [28], which in our model was restored in the Soleus muscle (see Figure 2A). Produced in muscle, BDNF regulates local translation in the junctional region, prevents atrophy of myofibers, and preserves axon myelination by the activation of mTOR [93,94,95]. As we assumed that BDNF had a limited effect on the TA MNs, and its level in the muscle was low, all observed changes in that muscle would be related to signaling from the antagonistic muscle, or indirect influence by stimulation from TrkB positive interneurons. 

Terminal Schwann cells (TSC) could have also contribute to the BDNF-related maintenance of cholinergic transmission. TSCs, which surround the peripheral synapse, were shown to be regulated by BDNF [96], to express BDNF and TrkB receptors [97], to react to neurotransmission [98], and to participate in the remodeling of NMJs, following nerve injury by the guidance of axonal reinnervation [99]. A recent study by Harrison and Rafuse proved a differential, muscle fiber type-dependent response of TSC to MN pathology [100]. The observed decrease in the S100b RNA levels in both muscles could be interpreted as a result of reported motoneuron diseases and spinal injuries, as well as the de-differentiation of Schwann cells [101,102]. The de-differentiated phenotype does not express S100b, as demonstrated in *in vitro* experiments [103]. Taking into account the normalized expression of the M2 receptor in the Soleus muscle after BDNF treatment, and in conjunction with recent findings that the M2 receptor is crucial for switching de-differentiated Schwann cells to the myelinating phenotype [59,104], we hypothesize that the BDNF treatment favors differentiation to regenerative instead of myelinating cells. The tightening of the structure and axonal guidance that characterized this SC phenotype partially explains the larger number of junctions that preserved pre-and postsynaptic contact in the BDNF Sol group.

The limitation of this study lies in the fact that the types of muscle fibers were not distinguished in the analysis; therefore, general conclusions are drawn with the assumption that the TA was fast and the Sol was slow muscle. The response of NMJs on slow-oxidative, fast-oxidative, and fast-glycolytic fibers is a key area that needs to be explored further, as well as spatial gene expression in the tripartite peripheral synapse. Enriching the TA MNs with TrkB receptors in this model could lead to beneficial changes, as BDNF improves axonal transport in fast MNs [105], and regulates energy homeostasis by phosphorylation of the AMP-activated protein kinase (AMPK) and acetylCoA carboxylase [106,107]. Future studies should aim to establish BDNF/TrkB treatment that targets selective pools of motor neurons.

## Figures and Tables

**Figure 1 biomedicines-10-02851-f001:**
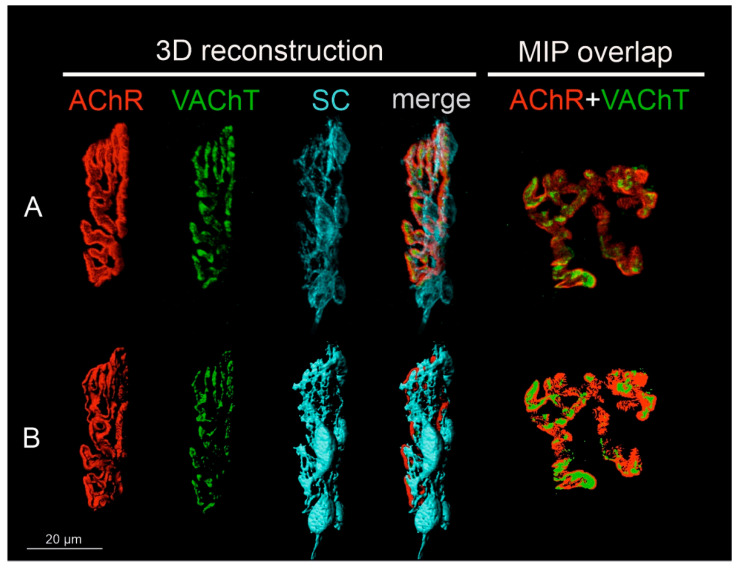
Steps of image preparation for the analysis of pre- and postsynaptic components of NMJs. All NMJs were subjected to 3D reconstruction. NMJs captured in en-face projection were analyzed additionally in 2D and shown as maximal intensity projection images (MIP). (**A**) Representative confocal photomicrograph of NMJ components of the Soleus muscle of the control rat: single channels that detected the fluorescence of α-bungarotoxin (BTx) bound to the nicotinic acetylcholine receptor (nAChR), immunofluorescence of the vesicular ACh transporter (VAChT), Schwann cells (SC), and a merge of them. The last image presents another control NMJ from the Soleus muscle, shown en-face, in maximal projection of the z-stack for VAChT and nAChR markers. All images were deconvoluted using the same parameters and randomly coded. (**B**) Three-dimensional masks of the fluorescence signal created with the use of Imaris software to measure volume and area of the structures. The last image presents 2D masks generated to measure the overlap of pre-and postsynaptic parts of the NMJ in ImageJ software.

**Figure 2 biomedicines-10-02851-f002:**
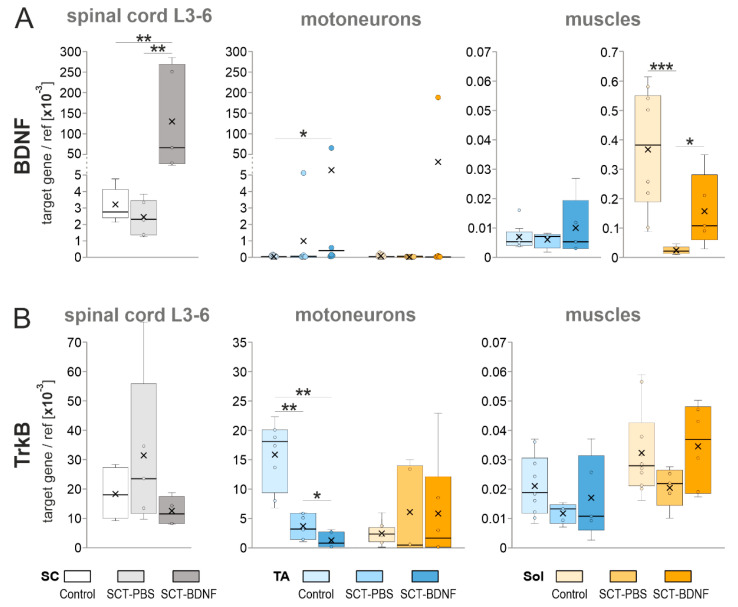
The effect of AAV-BDNF injection into the lumbar L1-2 spinal segment on (**A**) BDNF mRNA and (**B**) TrkB mRNA levels in the L3-6 spinal segments, TA, and Sol motoneurons and muscles 2 weeks after spinal cord transection. BDNF and TrkB mRNA levels were measured with a qPCR assay. Box and whisker plots show the minimum and maximum score (whiskers), first and third quartile (box), median (line), and mean (×). Data are from 5 Control, 6 SCT-PBS, and 5 SCT-BDNF rats (L3-6 segments); 8–9 Control, 7 SCT-PBS, and 6 SCT-BDNF rats (motoneurons); 9–10 Control, 4–6 SCT-PBS, and 5 SCT-BDNF rats (muscles). The Mann–Whitney *U*-test was used to assess differences between experimental groups (* *p* ≤ 0.05, ** *p* ≤ 0.01, *** *p* ≤ 0.001).

**Figure 3 biomedicines-10-02851-f003:**
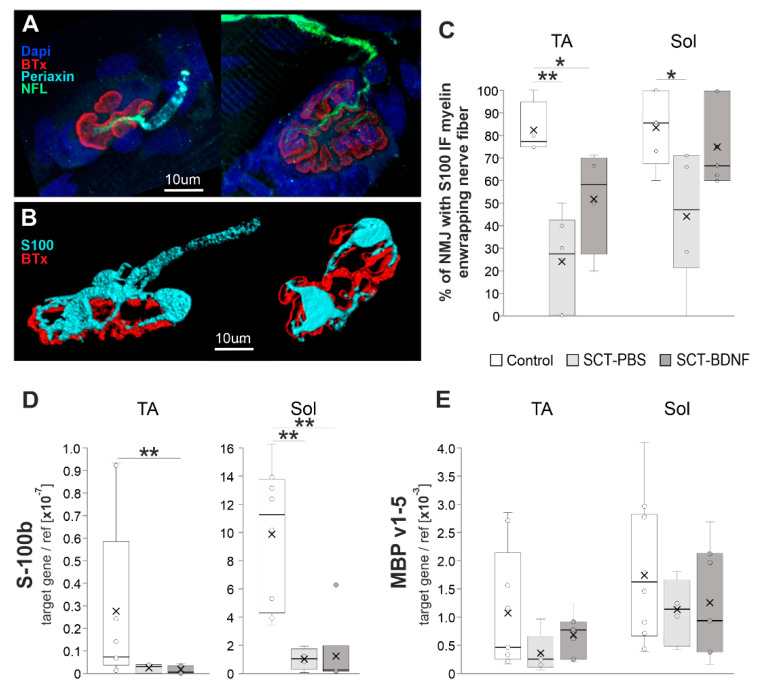
The effect of spinal cord transection and BDNF treatment on NMJ integrity. (**A**) Maximal projection of confocal images of TA NMJs showing a course of the nerve fibers identified with neurofilament, IF (NFL; green), and an arrangement of myelinating Schwann cells (SC, identified with periaxin; cyan). In the control rat (left), the branching motor axons that penetrated the junction folds (folds identified with fluorescent tagged alpha-bungarotoxin (BTx, red) bound to the nicotinic acetylcholine receptor (nAChR)), were wrapped in myelinating SC, while in SCT-PBS rat (right) the residual signal from myelinating SC was seen on the motor axon (see Appendix A for complete panel of images). (**B**) Three-dimensional reconstructions of the NMJ showing Schwann cells visualized with S-100 IF (cyan). In the Control NMJ (left), the myelinated nerve was connected to the endplate, however in the NMJ from SCT-PBS group (right), the myelinated nerve was not visible, although the non-myelinating terminal SCs (tSCs) still abutted the endplate. (**C**) Graph showing the percentage of nerves with preserved S100 IF-NMJ connections in the Control and SCT groups. Box and whisker plots show the minimum and maximum score (whiskers), first and third quartile (box), median (line), and mean (×). The number of NMJs analyzed in TA muscle: Control—17, SCT-PBS—37, SCT-BDNF—25; in the Sol muscle: Control—20, SCT-PBS—39, SCT-BDNF—39. (**D**,**E**) The effect of the SCT and AAV-BDNF treatment on (**D**) S100 mRNA and (**E**) MBP mRNA levels (qPCR assay) in the TA and Sol muscles. A profound decrease in S100b transcripts was observed in both muscles in spinal animals. mRNA data are from 10 Control, 4–5 SCT-PBS and 7 SCT-BDNF rats. The Mann–Whitney *U*-test was used (* *p* ≤ 0.05, ** *p* ≤ 0.01) to assess differences between experimental groups.

**Figure 4 biomedicines-10-02851-f004:**
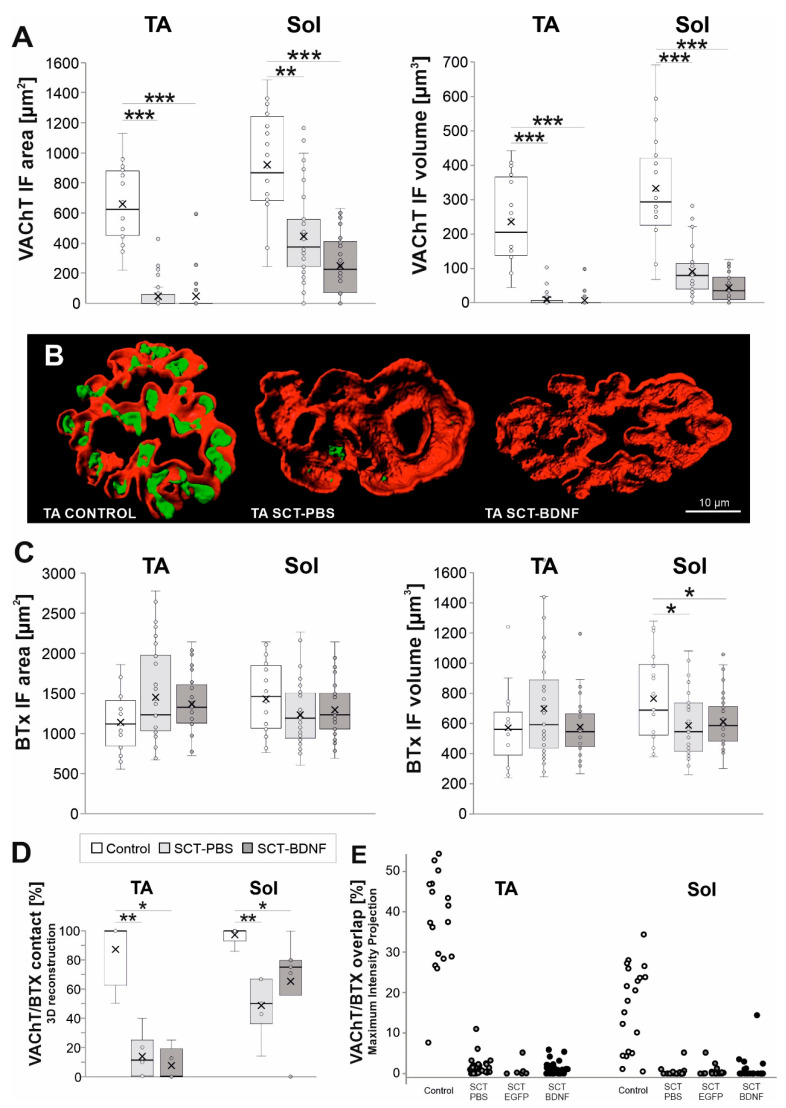
The effect of SCT and spinal BDNF overexpression on the VAChT IF and BTx signal at the NMJs of the Sol and TA muscles 2 weeks after surgery. (**A**) Changes in the area and volume of VAChT IF terminal boutons in 3D reconstructions of NMJs exemplified in (**B**) for TA NMJs. Box and whisker plots show the minimum and maximum score (whiskers), first and third quartile, median (line), and mean (×). Data are from 9–10 Control, 4–6 SCT-PBS, and 5 SCT-BDNF rats. (**B**) Comparison of the images of representative TA NMJs from the Control, SCT-PBS, and SCT-BDNF rat. Three-dimensional masks of the VAChT IF (green) and BTx (red) signals were created using the Imaris software. Films demonstrating 3D reconstructions of representative Sol NMJs from these three groups are available in the Appendix A. (**C**) Changes in the area and volume of BTx-binding sites of nicotinic ACh receptors located in the postsynaptic membrane in 3D reconstructions of NMJs. (**D**) Changes in NMJ integrity evaluated by VAChT adjacency to the endplate on side-views of 3D reconstructions of NMJs. The number of analyzed NMJs: TA: Control—17, SCT-PBS—37, SCT-BDNF—25, Sol: Control—20, SCT-PBS—39, SCT-BDNF—39. (**E**) VAChT/BTx-nAChR labeling overlap in the en-face images; Percent of overlapping calculated with the use of maximal intensity projection analysis. The number of analyzed NMJs: TA: Control—17, SCT-PBS—29, SCT-EGFP—17, SCT-BDNF—22, Sol: Control—17, SCT-PBS—30, SCT-EGFP—6, SCT-BDNF—32. The Mann–Whitney *U*-test was used to assess differences between experimental groups (* *p* ≤ 0.05, ** *p* ≤ 0.01, *** *p* ≤ 0.001).

**Figure 5 biomedicines-10-02851-f005:**
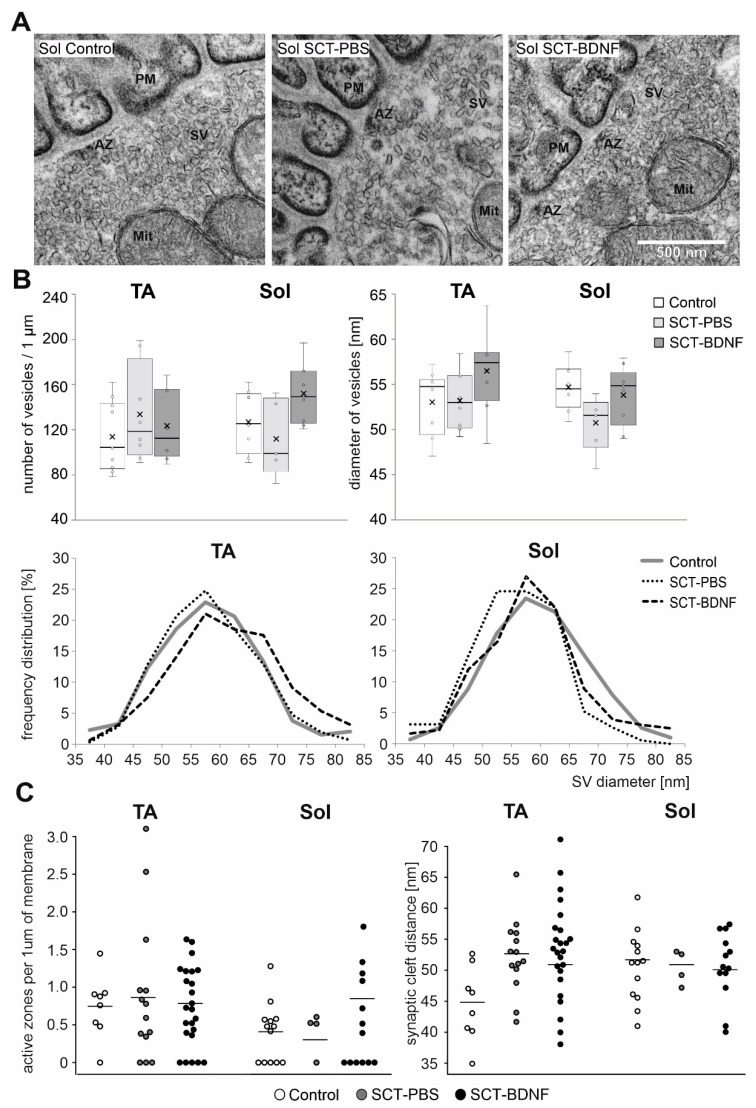
Comparison of synaptic vesicle distribution and density in motoneuron nerve terminals between Control and SCT groups. (**A**) Representative TEM micrographs of NMJs with maintained synaptic contacts from the Sol muscle of the Control, SCT-PBS, and SCT-BDNF rats. Scale bar—500 nm. SV—synaptic vesicles; PM—postsynaptic membrane. (**B**) Synaptic vesicle density in synaptic boutons in NMJs from the Control (8–10 images), SCT-PBS (5–8 images), and SCT-BDNF (8–9 images) groups. Lower panel shows profiles of the frequency distribution of SV pools varying in diameter [nm] in the same images of terminals of TA and Sol MNs, which were analyzed in (**A**). (**C**) Comparison of the number of active zones (AZ) per 1 µm of presynaptic membrane and synaptic cleft distance between groups. The points present data from individual images from the Control (8–14 images), SCT-PBS (4–14 images), and SCT-BDNF groups (13–24 images).

**Figure 6 biomedicines-10-02851-f006:**
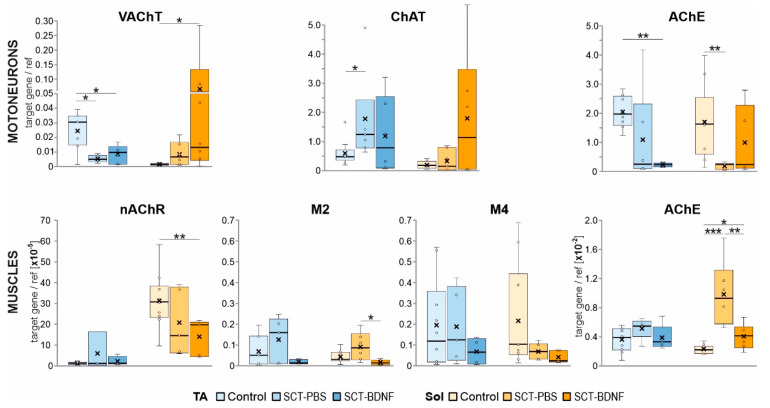
Effect of SCT and BDNF overexpression on mRNA levels of ACh vesicular transporter (VAChT), cholinergic enzymes (ChAT, AChE), muscarinic receptor (mAChR) subtypes, and nicotinic ACh receptor alpha−1 subunit (nAChR) in motoneurons (MNs) and muscles. **Upper panel**: Expression of the VAChT, ChAT, and AChE in the MNs. Data are from 9 Control, 4−6 SCT−PBS, and 5−6 SCT−BDNF rats. **Lower panel:** Expression of nAChR, M2, M4, mAChRs, and AChE in the muscle tissue. Data are from 8−10 Control, 4−6 SCT−PBS, and 4−6 SCT−BDNF rats. Box and whisker plots show the minimum and maximum score (whiskers), first and third quartile, median (line), and mean (×). The Mann–Whitney *U*−test was used to assess differences between experimental groups (* *p* ≤ 0.05, ** *p* ≤ 0.01, *** *p* ≤ 0.001).

**Figure 7 biomedicines-10-02851-f007:**
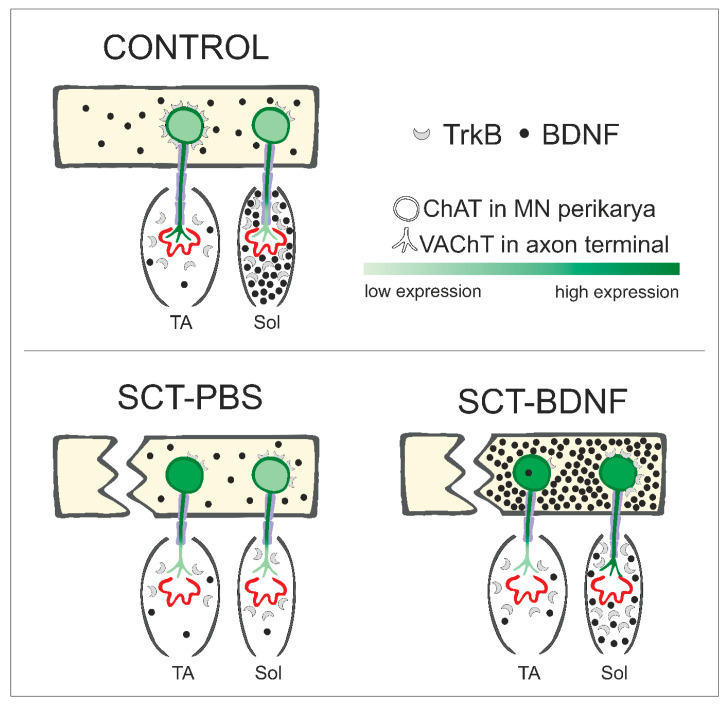
Schematic of changes in levels of BDNF, its TrkB receptor, and cholinergic marker expression in motor circuits of the ankle extensor (Sol) and flexor (TA) muscles under control conditions, SCT + PBS intraspinal injection, and SCT followed by intraspinal injection of the AAV-BDNF construct. TrkB receptors in the MNs are shown in the plasma membrane, assuming that the level of transcripts translated into receptor abundance on the cell surface.

**Table 1 biomedicines-10-02851-t001:** Number of animals in experimental groups.

Experimental Procedure/Tissue Preparation	CONTROL	SCT-PBS	SCT-EGFP	SCT-BDNF
Immunofluorescence (IF)/muscle fibers qPCR/spinal cord tissue qPCR/muscle tissue	5	6	4	7
Transmission Electron Microscopy (TEM)/muscle fibers	3	3	0	3
Laser Microdissection (LMD) followed by qPCR/motoneurons (MNs)	9	7	0	6

## Data Availability

Data supporting the reported results can be found in the Appendix A.

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
