# Peer review of "BDNF Spinal Overexpression after Spinal Cord Injury Partially Protects Soleus Neuromuscular Junction from Disintegration, Increasing VAChT and AChE Transcripts in Soleus but Not Tibialis Anterior Motoneurons"

_biomedicines, 2022, doi:10.3390/biomedicines10112851_

Round 1

Reviewer 1 Report (Previous Reviewer 1)

Review-2

The revised manuscript by Glowacka et al. responded to some of the points I highlighted and is now improved on the original version. New data implementing the study and/or arguments explaining the reasons for non-analysis have been provided. However, improvements in two aspects would still be welcome.

- I realize that the quantification of S100b marking would require a lot of effort, however the simple quantification of S100beta, MPZ, MBP transcripts by RT-qPCR would answer the question.

The manuscript still has many errors in English syntax and grammar, particularly in verb tenses. Although I have previously advised a small check, there has been no improvement in spelling. Therefore, I strongly recommend a revision by a native English speaker for correction.

Author Response

Reviewer 2 Report (Previous Reviewer 2)

The revised manuscript by Głowacka et al. is significantly improved and it provides valuable research findings in the field of BDNF effects in spinal cord injury.

Author Response

Reviewer 3 Report (Previous Reviewer 3)

I have carefully read the authors’ rebuttal letter. In my opinion, the authors have satisfactorily addressed some of the points that I have raised in the original reviewing. However, other important concerns still need to be corrected. Regarding the two main points I raised:

Point 1: In their rebuttal, the authors mention that they “made a group of 4 animals expressing EGFP” and that “they did not differ from SCT-PBS rats in locomotor abilities, post-surgery healing process or macroscopic tissue morphology”. I am still not convinced of the use of PBS instead of the sham AAV as control, and the fact that they have used it in previous publications is not something that I have to review. For this manuscript to be considered for publication, the authors must include in the article the data of the experiments comparing animals expressing EGFP with the ones using PBS as control. Also, in the Discussion section of the manuscript, the authors should properly disclose this sensitive issue in the same terms used in the rebuttal letter.

Point 2: My personal opinion regarding the use of the word “tendency” is not relevant. In subjective terms, there is a wide agreement in the scientific community that such description gives a misleading impression of the data, and tends to undermine the basic principles of accurate science communication. Hence, authors must completely avoid the use of the words “tendency” or “trend” throughout their manuscript.

This also applies for Point 8, as p<0.05 is considered as the minimum difference to be significant. Basically, there is no limit to the word “close” in the sentence: “However, we would like to discuss with the Reviewer a value of showing a p close to the threshold…”?, as in statistical terms, anything higher than 0.05 is not significant. I don't think there is something to discuss. Please also avoid the use of # to suggest differences, as they are not.

Quantification of the apposition ratio between pre- and postsynaptic areas is the correct way to evaluate the degree of denervation of the NMJ and how CST-BDNF could favor synaptic contacts of both cellular components. For this purpose, the same images obtained for the calculations in Figure 4 can be measured using ImageJ software, as indicated in the literature (Open Biol, 2016; 6(12)), and as it extensively used and accepted in the field.

In the TEM experiments depicted in Figure 5, the chosen images are not representative of the graphs. Please consider replacing them. Please quantify parameters such as active zones and synaptic cleft distance, as they will complement the idea that SCT-BDNF favors NMJ reinnervation. In this regard, also please integrate the results found in Figures 4 and 5 in the Discussion section. Also in the Discussion section, please describe why SCT gives rise to opposite effects on TrkB expression in motor neurons of the Soleus and Tibialis anterior muscles.

Round 2

Reviewer 3 Report (Previous Reviewer 3)

The authors have addressed most of my suggestions.

This manuscript is a resubmission of an earlier submission. The following is a list of the peer review reports and author responses from that submission.

Round 1

Reviewer 1 Report

The study of Glowacka et al. provides some interesting findings on the effects of BDNF overexpression in injured spinal cord upon spinal cord transection (SCT) on two different hindlimb muscles. Introduction gives extensive background of the processes induced by the injury and of the interest of BDNF overexpression by AAV-based transgene injection. 

However, title must be modified and the abstract should better point out the major results of the study. A more detailed description of AAV-BDNF construct would be also appreciable in materials and methods section, even if already published, and the age of the animals should be clearly indicated.

In the "results" section:

- The analysis of the protein expression of TrkB, including quantification, in the different tissues would be of interest (Fig 2).

-As the analysis of NMJ integrity represents a key results in the manuscripts, representative images of NMJ and 3D reconstructions should be included for all the conditions and for both muscles analyzed, and the quantitative expression of S100B and other Schwann cells markers should be alos showed (Fig 3).

-Images in Fig 4 should also represent all conditions

-Vesicles content might be quantified by protein expression of a proper marker (Fig 5)

-protein expression should be showed for ChAT, AchE, VAChT, mAChR, nAChR (Fig 6)

-In all figures legends definitions of groups/treatments should be homogeneous.

Reviewer 2 Report

The manuscript by Głowacka et al examines the interplay between the BDNF-TrkB system and its results upon overexpession in muscles and neuromuscular junctions, while it involves the cholinergic component as a potential mediator.

The study is strongly based in previous results from the research team and it now investigates the neurotrophin BDNF and its high affinity receptor in the neuronal and muscular parts after transection of the spinal cord. Although, the methodology is broad and clearly described, the results could be improved and enriched, in order to justify the conclusions. More specifically:

1. The authors have measured the levels of BDNF and its TrkB receptor in the gene level. What about their protein expression ? Is the TrkB receptor phosphorylated/activated upon overexpression of the BDNF?

2. What about the expression of other neurotrophic components (GDNF, NGF, NT-3) of the examined tissue before and after the transection? If the low-affinity panneurotrophin receptor p75 is also expressed post-synaptically, it could also mediate -partially at least- some of the neurotrophic effects. p75NTR is known to be upregulated after injury in several neuronal tissues.

3. Although, the 3D images offer significant information, the authors could also show some more detailed cellular signals, especially when specific markers are used. Co-staining with 2 or 3 markers could provide significant data for better interpretation of the results.

4. Some results need further examination and explanation. How is it explained that motor neurons have low expression of BDNF and TrkB, even upon the BDNF overexpression? This neuronal type is highly dependent in BDNF trophic support. Also, the lack of any change at synaptic vesicles needs better justification.

Reviewer 3 Report

The manuscript entitled “Effects of BDNF spinal overexpression on morphological and functional correlates of pre- and postsynaptic components of cholinergic signaling of neuromuscular junction in tibialis anterior and soleus muscles after spinal cord injury” by GÅ‚owacka et al. aimed to determine whether intraspinal AAV-BDNF treatment of rats with spinal cord transection (SCT) can counteract the postlesion changes in cholinergic signaling throughout the motor unit.

The manuscript contains potentially interesting findings. However, the lack of some basic features of the experimental design and the way the results are interpreted precludes any enthusiasm to give a good evaluation of this paper.

Specifically, my main concerns are: 1- the use of PBS as a control of an AAV to overexpress BDNF (instead of the control (sham) AAV), and 2- data analysis (e.g. only some animals showed the expected results) and interpretation, particularly the use of “tendency” on data that are not statistically significant. How do authors define “tendency”?

Only if authors could satisfactorily fulfill these two essential conditions, some other specific comments to the manuscript are:

1- In biological terms, how is it possible to interpret a finding like this: “AAV- 332 BDNF administration resulted also in high BDNF mRNA levels in TA MNs of 4 rats, and in Sol MNs of 2 rats; in the remaining rats no BDNF overexpression was found”.

Also in this regard, was at least the infection efficiency evaluated through, for instance, the expression of a tagged protein?

Also, regarding the subsequent data in the paper,  was it built with data from the 9 animals? or only those expressing BDNF?

2- The “box and whisker plots show the minimum and maximum score (whiskers), first and third quartile (box), median (line), and mean (x)” is a very complex and seemingly forced way of presenting the data.

In my opinion, a bar + dots (one dot per experimental n) would be a simpler way of presenting the results.

Also, what was the data considered to assess the statistical differences? Raw data? Means? Medians? Why and for which set of data the Wilcoxon or Mann-Whitney U tests were used?

3- Why # p=0.075 is mentioned? To denote tendency?

4- The word “motor unit” is erroneously used throughout the manuscript.

5- Why do the images in Figs 3 and 4 not include the SCT-BDNF condition?

6- Why when assessing if VAChT signal (presynaptic) is in apposition to nAChR-associated Btx signal the signal abundance was ignored?  If this is a measure of denervation (total or partial), the percentage of apposition is critical. A ratio between pre and postsynaptic areas is routinely used in the field and will give an accurate measurement, rather than a blind qualitative assessment.

7- In lines 465-466 “In the Sol muscle, a higher density of SV was observed in some terminals of SCT-BDNF rats compared to SCT- PBS rats”.

What do authors mean with “some”? What does this mean? How could this be useful?

8- Regarding experiments in Fig. 5, how do authors know that the MNs analyzed are the ones innervating the TA or Soleus muscles?

9- There’s no mention of the well-known expression of muscarinic AChRs in perisynaptic terminal Schwann cells.

10- Instead of going through each result, the discussion section should be organized in an integrative fashion.

Round 2

Reviewer 2 Report

The authors of the revised manuscript entitled "BDNF overexpression after spinal cord injury partially protects Soleus neuromascular junction from disintegration, upregulating VAChT-AChE transcripts in Soleus but not Tibialis anterior motor neurons" have clarified all the important questions and the overall quality of the research has been significantly improved.

The only minor suggestion is to add information of the used antibodies (which antibody is depicted for each color) in Supplementary Figure 4.